# Causes of death in children with congenital Zika syndrome in Brazil, 2015 to 2018: A nationwide record linkage study

**Maria da Conceição N. Costa**[1,2☯], **Luciana Lobato Cardim**[1☯], **Cynthia A. Moore**[3], **Eliene dos Santos de Jesus**[2,4], **Rita Carvalho-Sauer**[2,5], **Mauricio L. Barreto**[1,2], **Laura C. Rodrigues**[1,6], **Liam Smeeth**[6], **Lavínia Schuler-Faccini**[7], **Elizabeth B. Brickley**[6], **Wanderson K. Oliveira**[8], **Eduardo Hage Carmo**[1,9], **Julia Moreira Pescarini**[1,6], **Roberto F. S. Andrade**[1,10], **Moreno M. S. Rodrigues**[1], **Rafael V. Veiga**[1], **Larissa C. Costa**[1], **Giovanny V. A. França**[9], **Maria Gloria Teixeira**[1,2‡], **Enny S. Paixão**[1,6‡*]

**1** Center of Data and Knowledge Integration for Health (CIDACS), Gonçalo Moniz Institute, Oswaldo Cruz Foundation, Salvador, Bahia, Brazil, **2** Collective Health Institute, Federal University of Bahia, Salvador, Bahia, Brazil, **3** Goldbelt Professional Services, LLC, Chesapeake, Virginia, United States of America, **4** Municipal Health Department, Department of Health Information, Salvador, Bahia, Brazil, **5** East Regional Health Center, State Health Secretariat of Bahia, Santo Antonio de Jesus, Bahia, Brazil, **6** London School of Hygiene and Tropical Medicine, London, United Kingdom, **7** Department of Genetics, Federal University of Rio Grande do Sul, Porto Alegre, Rio Grande do Sul, Brazil, **8** Technical Directorate of Education and Research, Ministry of Defense Hospital das Armed Forces, Brasília, Distrito Federal, Brazil, **9** Secretariat of Health Surveillance, Ministry of Health, Brasilia, Distrito Federal, Brazil, **10** Physics Institute, Federal University of Bahia, Salvador, Bahia, Brazil

☯ These authors contributed equally to this work.
‡ MGT and ESP also contributed equally to this work.
* enny.cruz@lshtm.ac.uk

**Data Availability Statement:** All data supporting the findings presented here were obtained from Centro de Integração de Dados e Conhecimentos

## Abstract

### Background

Children with congenital Zika syndrome (CZS) have severe damage to the peripheral and central nervous system (CNS), greatly increasing the risk of death. However, there is no information on the sequence of the underlying, intermediate, immediate, and contributing causes of deaths among these children. The aims of this study are describe the sequence of events leading to death of children with CZS up to 36 months of age and their probability of dying from a given cause, 2015 to 2018.

### Methods and findings

In a population-based study, we linked administrative data on live births, deaths, and cases of children with CZS from the SINASC (Live Birth Information System), the SIM (Mortality Information System), and the RESP (Public Health Event Records), respectively. Confirmed and probable cases of CZS were those that met the criteria established by the Brazilian Ministry of Health. The information on causes of death was collected from death certificates (DCs) using the World Health Organization (WHO) DC template. We estimated proportional mortality (PM%) among children with CZS and among children with non-Zika CNS congenital anomalies (CA) by 36 months of age and proportional mortality ratio by cause (PMRc). A

para Saúde (CIDACS). Importantly, restrictions apply to the availability of these data. However, upon reasonable request and provided all ethical and legal requirements are met, the institutional data curation team can make the data available. Information on how to apply to access the data can be found at https://cidacs.bahia.fiocruz.br/.

**Funding:** This work was supported by the Secretary of Health Surveillance, Ministry of Health of Brazil, and by grants from the Wellcome Trust (213589/Z/18/Z, to ESP); (205377/Z/16/Z, to MLB), the European Union Horizon 2020 Research and Innovation Program under the Zika Preparedness Latin American Network (ZikaPLAN; 734584, to EBB). The funders had no role in the study design, analysis, decision to publish or preparation of the manuscript.

**Competing interests:** The authors have declared that no competing interests exist.

**Abbreviations:** CA, congenital anomalies; CNS, central nervous system; CZS, congenital Zika syndrome; MoH, Ministry of Health; PM, proportional mortality; PMRc, proportional mortality ratio by cause; WHO, World Health Organization; ZIKV, Zika virus.

total of 403 children with confirmed and probable CZS who died up to 36 months of age were included in the study; 81.9% were younger than 12 months of age. Multiple congenital malformations not classified elsewhere, and septicemia unspecified, with 18 (PM = 4.5%) and 17 (PM = 4.2%) deaths, respectively, were the most attested underlying causes of death. Unspecified septicemia (29 deaths and PM = 11.2%) and newborn respiratory failure (40 deaths and PM = 12.1%) were, respectively, the predominant intermediate and immediate causes of death. Fetuses and newborns affected by the mother's infectious and parasitic diseases, unspecified cerebral palsy, and unspecified severe protein-caloric malnutrition were the underlying causes with the greatest probability of death in children with CZS (PMRc from 10.0 to 17.0) when compared to the group born with non-Zika CNS anomalies. Among the intermediate and immediate causes of death, pneumonitis due to food or vomiting and unspecified seizures (PMRc = 9.5, each) and unspecified bronchopneumonia (PMRc = 5.0) were notable. As contributing causes, fetus and newborn affected by the mother's infectious and parasitic diseases (PMRc = 7.3), unspecified cerebral palsy, and newborn seizures (PMRc = 4.5, each) were more likely to lead to death in children with CZS than in the comparison group. The main limitations of this study were the use of a secondary database without additional clinical information and potential misclassification of cases and controls.

## Conclusion

The sequence of causes and circumstances involved in the deaths of the children with CZS highlights the greater vulnerability of these children to infectious and respiratory conditions compared to children with abnormalities of the CNS not related to Zika.

## Author summary

### Why was this study done?

- Children with congenital Zika syndrome (CZS) have a broad spectrum of clinical manifestations due to sequelae of central and peripheral nervous system damage that can impair important vital functions and greatly increase the risk of death in affected children.

- Little is known about the circumstances involved in the death of children affected by this syndrome.

- Analysis of the sequences of causes attested on the death certificate (DC) can contribute to this understanding and thus support the implementation of health programs and specific postnatal protocols to meet the needs of these children.

- Brazil was the country with the highest number of live births with CZS.

### What did the researchers do and find?

- We linked administrative data of all reporting live births, deaths, and notification of probable and confirmed CZS cases in Brazil in the period from January 1, 2015 to December 31, 2018.

- We analyzed the proportional mortality of the main underlying, intermediate, immediate, and contributing causes of death of 403 children with CZS up to 36 months age and compared with the same causes of death of 734 children with non-Zika–related CNS congenital anomalies (CA) who died by 36 months of age from the proportional mortality ratio.

- The leading underlying causes of death for children with CZS were multiple congenital malformations not classified elsewhere and unspecified septicemia.

### What do these findings mean?

- The sequence of causes and circumstances involved in the deaths of the CZS children by 36 months age, in Brazil, highlighted the greater vulnerability of these children to infectious and respiratory conditions.

- It is necessary to develop evidence-based health protocols and raising awareness of conditions associated with high mortality in children affected by congenital Zika virus infection to guide health professionals responsible for caring for these children.

## Introduction

Intrauterine infection with Zika virus (ZIKV) has the potential to cause a broad spectrum of clinical manifestations. Based on current knowledge, most problems are sequelae of central and peripheral nervous system damage [1] that can impair important vital functions and greatly increase the risk of early death in affected children with and without microcephaly [2,3]. Previous studies have shown that the main factor associated with deaths among children with congenital Zika syndrome (CZS) were very low birth weight and low Apgar score [4].

Despite the evolution of knowledge about CZS, including clinical manifestations [5], case fatality [4] and mortality rates [2], and risk factors for death [2], little is known about the circumstances involved in the death of children affected by this syndrome. Analysis of the sequences of causes attested on the death certificate (DC) can contribute to this understanding and supports the implementation of health programs and specific postnatal protocols to meet the needs of affected children.

This study analyzes the morbid conditions that made up the sequence of events leading to a fatal outcome in children up to 36 months of age, from January 1, 2015 to December 31, 2018, in Brazil, the country with the highest number of live births with CZS. More specifically, the aims of this study are the following: (i) identify the main reported underlying, intermediate, immediate, and contributing causes of death of children with CZS; and (ii) analyze the probability of each of these causes of death occurring among children with CZS when compared to a

group of children with central nervous system (CNS) anomalies not related to congenital ZIKV infection.

## Methods

We developed a retrospective population-based study of mortality in children with CZS, linking administrative data reporting live births, deaths, and notification of CZS cases in Brazil, in the period from January 1, 2015 to December 31, 2018.

A written analysis plan was not developed; however, all analyses were presented and discussed during regular study group meetings. This study is reported as per the Reporting of studies Conducted using Observational Routinely collected health Data (RECORD) Statement (S1 RECORD checklist) [6].

### Data sources

We obtained data from the Sistema de Informação sobre Nascidos Vivos—SINASC (live birth information system), the Registro de Eventos em Saúde Pública—RESP (public health event records), and the Sistema de Informação sobre Mortalidade—SIM (mortality information system).

SINASC is updated with the mandatory registration of all live births in Brazil. The standardized data collection instrument is the Declaração de Nascido Vivo (declaration of a live birth), which is completed by the health professional who assisted in the delivery. In 2013, SINASC coverage was already 100% in most Brazilian states [7]. From SINASC, we retained information on the date of birth and the respective International Classification of Diseases 10th revision (ICD-10) codes for congenital abnormalities [8].

RESP [9] is an online form developed by the Brazilian Ministry of Health (MoH) and used by all health services in Brazil to notify suspected cases of congenital abnormalities related to ZIKV infection. From RESP, we obtained information on the final classification of cases (whether confirmed, probable, discarded, or under investigation) [10].

SIM is updated with the registration of mortality data. The DC issuance is mandatory for all deaths occurring in the country. In 2011, this system covered an estimated 96.1% of all deaths of residents in Brazil [11]. From SIM, we obtained the ICD-10 of causes of death, age, and date of death.

The section of the DC completed with the cause of death meets the "International Model of the Medical Certificate of Cause of Death" recommended by the World Health Assembly in 1948 that is filled out exclusively by physicians. The causes of death to be recorded on the DC are all morbid conditions or injuries that produced or contributed to death, as well the circumstances that produced these injuries. In part I, the physicians must declare the causes that directly led the death, in the following order: immediate or terminal cause of death in the first line (a), intermediate causes in subsequent lines (b and c), and in the last line (d), the underlying cause (the one which started the succession of morbid events or circumstances that led directly to death). In part II, other important preexisting morbid conditions that may have indirectly contributed to the death but did not enter the causal sequence stated in part I, are reported (S1 Fig) [12].

The causes of death attested by the physician subsequently receive an ICD-10 code by qualified technicians from the Municipal Health Department, and then the data from the DC are included in SIM. SIM has an automatic selector capable of identifying and reclassifying the underlying cause, regardless of the line in which it was recorded on the DC and termed "revised underlying cause." Generally, the underlying cause is used for the production of mortality statistics for the country and to define public policies.

According to the guidelines of the Brazilian MoH [13] if, during the coding process, the underlying cause attested by the physician on the DC is considered an uninformative code for health statistics (for example, ill-defined causes belonging to chapter XVIII of the ICD-10), it must be investigated. In Brazil, this procedure is performed by the local Epidemiological Surveillance team using a specific form in which all clinical manifestations are registered and then the sequence of events (diseases or injury) that directly led to the death are redefined by another certifying physician from the Municipal Health Department. Then, the new causes of death are coded with the appropriate ICD-10 and included in SIM, while the cause recorded on the original DC is preserved.

S2 Fig shows that the variable "revised underlying cause," created by SIM after automated reclassification, was considered the underlying cause of death. If only 1 cause was listed on the DC, it was considered as the underlying cause. Intermediate and immediate causes were determined by a schema based on completed lines of data in the SIM. Contributing causes were all causes recorded in part II of the DC. After applying the conditionals mentioned in this figure for the intermediate, immediate, and contributing causes, we manually excluded all the causes from other lines that were considered an underlying cause after review by the automated underlying cause selector of SIM.

## Linkage process

All children with information recorded in the SINASC were eligible for linkage with RESP and SIM. The Centro de Integração de Dados e Conhecimentos para a Saúde—CIDACS (Center of Integration of Data and Knowledge for Health) performed this procedure using a non-deterministic record linkage tool (CIDACS-RL) specifically designed to conduct large-scale observational studies with administrative data from Brazil [14].

## Case definition

We included all children whose death occurred by 36 months of age and met criteria to be included in one of the following 2 groups (S3 Fig):

**Children with CZS.** According to criteria established by the Brazilian MoH, children linked with RESP and classified as confirmed cases had signs and symptoms compatible with the syndrome and either laboratory testing consistent with congenital ZIKV infection or maternal report of fever or rash during pregnancy or both.

Those classified as probable CZS cases had a clinical description compatible with CZS, negative testing for other congenital infections although without specific testing for ZIKV infection available, and an asymptomatic mother during pregnancy [10].

**Children with non-Zika CNS congenital anomalies.** Children with a record in SINASC of congenital CNS anomalies non- Zika–related coded as Q01, Q05-Q07 (ICD-10) [15], who were not linked with RESP.

Suspected cases of CZS reported in the RESP that were under epidemiological investigation or were inconclusive were excluded from this study. Live births with ICD-10 code Q00 (anencephaly) were excluded from the comparison group.

## Data analysis

For both groups of children (with CZS and with non-Zika–related CNS congenital abnormalities), the proportional mortality—PM (%), for the period 2015 to 2018, was calculated for each ICD-10 group and types of causes of deaths defined by World Health Organization (WHO) and adopted by the Brazilian MoH [13]. PM (%) by a cause corresponds to the ratio between the number of deaths due to this cause and the total number of deaths, multiplied by 100.

The chance of children born with CZS dying from a given cause compared to children with CNS anomalies unrelated to ZIKV was estimated from the proportional mortality ratio by cause (PMRc). This corresponds to the ratio between the PM (%) of each cause (and each group of causes) of death for children with CZS and the PM (%) by the same cause estimated for children with non-Zika–related CNS congenital anomalies (CA).

We assessed the increase (fold increases) for selected cause of death of children born with CZS by aggregating deaths regardless of their position in the sequence of events and comparing with the number of deaths that would be attributed to each cause when relying only on the underlying condition.

Because of potential misclassification of children during the Zika epidemic, we performed a sensitivity analysis using as a comparison group children born before the Zika epidemic in Brazil, in 2012 to 2013, with record of CNS anomalies considered non-Zika–related (Q01, Q05-Q07—ICD-10). The results were presented in the Supporting information.

In Tables 1–4, we only present the causes that accounted for at least 3 deaths among children with CZS, while the remaining causes were classified as "other groups and attested causes for less than 3 deaths."

## Ethics statement

This study analyzed unidentified data and was approved by the Federal University of Bahia Institute of Health Collective Research Ethics Committee (Certificate of Presentation of Ethical Appreciation—CPEA number 70745617.2.0000.5030). No informed consent was required for this study.

## Results

Of the 1,110 live births in 2015 to 2018 reported in SINASC and linked to RESP and to SIM, 36.3% (403) of the children died up to 36 months of age, and were classified as confirmed (75.2%) and probable (24.8%) cases of CZS. Of these 403 deaths, 81.9% occurred among children under 1 year of age, of which 50.1% were less than 28 days old, and 76.6% of these were 0 to 6 days old. From 12 up to 36 months of age, 73 (18.1%) of the 403 deaths occurred. Regarding the children with non-Zika CNS anomalies recorded in SINASC, 66.0% (734/1,112) were included in this study. Of these 92.5% occurred among children under 1 year of age, of which 62.4% were less than 28 days old, and 77.1% of these were 0 to 6 days old.

### Main causes of death up to 36 months of age in children born with CZS

**Underlying causes.** The groups of causes of death that presented the highest frequencies of underlying causes of death recorded among the 403 children with CZS were the following: congenital malformations, deformities, and chromosomal anomalies; some conditions originating in the perinatal period; and some infectious and parasitic diseases with 232, 58, and 35 deaths, respectively. Microcephaly, multiple congenital malformations not classified elsewhere, and unspecified septicemia, with respectively 93 (PM = 23.1%), 18 (PM = 4.5%), and 17 (PM = 4.2%) deaths, were the conditions most recorded by the physicians as underlying causes of death. Microcephaly with PMRc = 57.8; fetus and newborn affected by infectious and parasitic diseases of the mother with PMRc = 17.0; unspecified cerebral palsy with PMRc = 12.0; and unspecified severe protein-calorie malnutrition with PMRc = 10.0 were the underlying causes that relatively contributed more to death in children with CZS than for those of the comparison group (Table 1).

**Intermediate causes.** Table 2 shows 259 recorded intermediate causes of death for children with CZS. The greatest frequencies of the causal groups were 67 for some conditions

**Table 1. Main underlying causes of death (number and proportional mortality/PM%) of 403 children up to 36 months of age, born with CZS, values of these indicators for those born with CA of the CNS non-Zika related and PMRc according to groups and types of causes[1]; Brazil, 2015–2018.**

| Groups and types of causes[1] | CZS | | CA of CNS non-Zika related | | PMRc |
|---|---|---|---|---|---|
| | N | PM(%) | N | PM(%) | |
| **Some infectious and parasitic diseases (A00—B99)** | **35** | **8.7** | **14** | **1.9** | **4.6** |
| A41.9—Unspecified septicemia | 17 | 4.2 | 12 | 1.6 | 2.6 |
| A92.8—Other specified viral fevers transmitted by mosquitoes | 6 | 1.5 | - | - | - |
| **Endocrine. nutritional and metabolic diseases (E00-E90)** | **7** | **1.7** | **1** | **0.1** | **17.0** |
| E43—Severe protein-calorie malnutrition unspecified | 4 | 1.0 | 1 | 0.1 | 10.0 |
| **Nervous system diseases (G00-G99)** | **20** | **5.0** | **26** | **3.5** | **1.4** |
| G80.9—Unspecified cerebral palsy | 5 | 1.2 | 1 | 0.1 | 12.0 |
| **Respiratory system diseases (J00-J99)** | **18** | **4.5** | **14** | **1.9** | **2.4** |
| J18.9—Unspecified pneumonia | 8 | 2.0 | 5 | 0.7 | 2.9 |
| **Some conditions originating in the perinatal period (P00-P96)** | **58** | **14.4** | **92** | **12.5** | **1.2** |
| P00.2—Fetus and newborn affected by the mother's infectious and parasitic diseases | 7 | 1.7 | 1 | 0.1 | 17.0 |
| P21.9—Asphyxia at birth unspecified | 4 | 1.0 | 2 | 0.3 | 3.3 |
| P22.0—Newborn respiratory distress syndrome | 3 | 0.7 | 7 | 1.0 | 0.7 |
| P36.9—Unspecified bacterial septicemia of the newborn | 11 | 2.7 | 10 | 1.4 | 1.9 |
| P37.1—Congenital toxoplasmosis | 5 | 1.2 | - | - | - |
| **Congenital malformations, deformities, and chromosomal anomalies (Q00-Q99)** | **232** | **57.6** | **552** | **75.2** | **0.8** |
| Q00.0—Anencephaly | 10 | 2.5 | 9 | 1.2 | 2.1 |
| Q01.9—Encephalocele unspecified | 3 | 0.7 | 73 | 9.9 | 0.1 |
| Q02—Microcephaly | 93 | 23.1 | 3 | 0.4 | 57.8 |
| Q03.1—Atresia of the Clefts of Luschka and the foramen of Magendie | 4 | 1.0 | 3 | 0.4 | 2.5 |
| Q03.9—Unspecified congenital hydrocephalus | 9 | 2.2 | 12 | 1.6 | 1.4 |
| Q04.2—Holoprosencephaly | 4 | 1.0 | 7 | 1.0 | 1.0 |
| Q04.3—Other deformities due to brain reduction | 12 | 3.0 | 7 | 1.0 | 3.0 |
| Q07.9—Unspecified congenital malformation of the nervous system | 6 | 1.5 | 5 | 0.7 | 2.1 |
| Q24.9—Unspecified malformation of the heart | 9 | 2.2 | 22 | 3.0 | 0.7 |
| Q33.6—Lung hypoplasia and dysplasia | 10 | 2.5 | 17 | 2.3 | 1.1 |
| Q89.7—Multiple congenital malformations not classified elsewhere | 18 | 4.5 | 40 | 5.4 | 0.8 |
| Q89.9—Multiple congenital malformations not classified elsewhere | 11 | 2.7 | 26 | 3.5 | 0.8 |
| **Symptoms. Abnormal signs and findings from clinical and laboratory examinations. Unclassified elsewhere (R00-R99)** | **7** | **1.7** | **6** | **0.8** | **2.1** |
| R99—Other ill-defined and unspecified causes of mortality | 5 | 1.2 | 2 | 0.3 | 4.0 |
| Other groups and types of causes | 26 | 6.5 | 29 | 4.0 | 1.6 |
| **Total** | **403** | **100.0** | **734** | **100.0** | **1.0** |

Source: Center of Data and Knowledge for Health-CIDACS: Linkage of the Live Birth Information System/SINASC. Public Health Events Registry/RESP and Mortality Information System/SIM.

[1]ICD 10 (International Classification of Diseases and Causes of Death (ICD 10th Revision)).

Only causes of death whose absolute frequency were ≥3 are included separately.

PM% calculated in relation to the total of causes of death.

CA, congenital anomalies; CNS, central nervous system; CZS, congenital Zika syndrome; PM, proportional mortality; PMRc, proportional mortality ratio by cause.

originating in the perinatal period; 50 for congenital malformations, chromosomal deformities, and anomalies; and 42 for respiratory system diseases. Unspecified septicemia, microcephaly, and unspecified pneumonia were reported to be the most frequent intermediate causes of death among children with CZS in our study (29, 23, and 12 recorded causes of

**Table 2. Main intermediate causes of death (number and proportional mortality/PM%) of 403 children up to 36 months of age, born with CZS, values of these indicators for those born with CA of the CNS non-Zika related and PMRc according to groups and types of causes[1]; Brazil, 2015–2018.**

| Groups and types of causes[1] | CZS | | CA of CNS non-Zika related | | PMRc |
|---|---|---|---|---|---|
| | N | PM(%) | N | PM(%) | |
| **Some infectious and parasitic diseases (A00—B99)** | **35** | **13.5** | **25** | **5.4** | **2.5** |
| A41.9—Unspecified septicemia | 29 | 11.2 | 21 | 4.5 | 2.5 |
| **Respiratory system diseases (J00-J99)** | **42** | **16.2** | **49** | **10.6** | **1.5** |
| J18.9—Unspecified pneumonia | 12 | 4.6 | 20 | 4.3 | 1.1 |
| J69.0—Pneumonitis due to food or vomiting | 5 | 1.9 | 1 | 0.2 | 9.5 |
| J96.0—Acute breathing insufficiency | 3 | 1.2 | 2 | 0.4 | 3.0 |
| J96.9—Unspecified respiratory failure | 9 | 3.5 | 4 | 0.9 | 3.9 |
| **Genitourinary system diseases (N00-N99)** | **5** | **1.9** | **14** | **3.0** | **0.6** |
| N17.9—Acute kidney failure unspecified | 3 | 1.2 | 2 | 0.4 | 3.0 |
| **Some conditions originating in the perinatal period (P00-P96)** | **67** | **25.9** | **167** | **36.0** | **0.7** |
| P07.1—Other low birth weight newborns | 8 | 3.1 | 12 | 2.6 | 1.2 |
| P07.2—Extreme immaturity | 3 | 1.2 | 6 | 1.3 | 0.9 |
| P21.9—Asphyxia at birth unspecified | 3 | 1.2 | 7 | 1.5 | 0.8 |
| P22.9—Unspecified newborn respiratory distress | 4 | 1.5 | 3 | 0.6 | 2.5 |
| P23.9—Unspecified congenital pneumonia | 5 | 1.9 | 3 | 0.6 | 3.2 |
| P28.5—Newborn respiratory failure | 4 | 1.5 | 17 | 3.7 | 0.4 |
| P36.9—Unspecified bacterial septicemia of the newborn | 5 | 1.9 | 25 | 5.4 | 0.4 |
| P39.9—Infection of the perinatal period not specified | 8 | 3.1 | 5 | 1.1 | 2.8 |
| **Congenital malformations, chromosomal deformities, and anomalies (Q00-Q99)** | **50** | **19.3** | **92** | **19.8** | **1.0** |
| Q02—Microcephaly | 23 | 8.9 | 2 | 0.4 | 22.3 |
| Q04.8—Other specified congenital malformations of the brain | 3 | 1.2 | - | - | - |
| Q07.9—Unspecified congenital malformation of the nervous system | 4 | 1.5 | 2 | 0.4 | 3.8 |
| Q89.9—Unspecified congenital malformations | 5 | 1.9 | 6 | 1.3 | 1.5 |
| **Abnormal symptoms, signs and findings from clinical and laboratory examinations, unclassified elsewhere (R00-R99)** | **16** | **6.2** | **14** | **3.0** | **2.1** |
| R56.8—Other and unspecified seizures | 5 | 1.9 | 1 | 0.2 | 9.5 |
| Other groups and types of causes | 44 | 17.0 | 103 | 22.2 | 0.8 |
| **Total** | **259** | **100.0** | **464** | **100.0** | **1.0** |

Source: Center of Data and Knowledge for Health-CIDACS: Linkage of the Live Birth Information System/SINASC. Public Health Events Registry/RESP and Mortality Information System/SIM.

[1]ICD 10 (International Classification of Diseases and Causes of Death (ICD 10th Revision)).

Only causes of death whose absolute frequency were ≥3 are included separately.

PM% calculated in relation to the total of causes of death.

CA, congenital anomalies; CNS, central nervous system; CZS, congenital Zika syndrome; PM, proportional mortality; PMRc, proportional mortality ratio by cause.

death, and PM of 11.2%, 8.9%, and 4.6%, respectively). Microcephaly with PMRc = 22.3 and pneumonitis due to food or vomiting and other and unspecified seizures, with PMRc = 9.5 each, were the recorded intermediate causes of death that contributed relatively more to deaths of children with CZS when compared to those with CNS congenital abnormalities, non-Zika related. This was followed by unspecified respiratory failure with PMRc = 3.9, unspecified congenital malformation of the nervous system with PMRc = 3.8, and unspecified congenital pneumonia with PMRc = 3.2.

**Immediate causes.** Table 3 shows 330 causes of death recorded as having been immediate or terminal for children with CZS. The causal groups some conditions originating in the

**Table 3. Main immediate causes of death (number and proportional mortality/PM%) of 403 children up to 36 months of age, born with CZS, values of these indicators for those born with CA of the CNS non-Zika related and PMRc according to groups and types of causes[1]; Brazil, 2015–2018.**

| Groups and types of causes[1] | CZS | | CA of CNS non-Zika related | | PMRc |
|---|---|---|---|---|---|
| | N | PM(%) | N | PM(%) | |
| **Some infectious and parasitic diseases (A00—B99)** | **41** | **12.4** | **79** | **13.7** | **0.9** |
| A41.9—Unspecified septicemia | 39 | 11.8 | 74 | 12.8 | 0.9 |
| **Circulatory system diseases (I00-I99)** | **7** | **2.1** | **6** | **1.0** | **2.1** |
| I50.9—Unspecified heart failure | 4 | 1.2 | 2 | 0.3 | 4.0 |
| **Respiratory system diseases (J00-J99)** | **63** | **19.1** | **55** | **9.5** | **2.0** |
| J18.0—Unspecified bronchopneumonia | 5 | 1.5 | 2 | 0.3 | 5.0 |
| J18.9—Unspecified bronchopneumonia | 6 | 1.8 | 4 | 0.7 | 2.6 |
| J80—Adult respiratory distress syndrome | 5 | 1.5 | 4 | 0.7 | 2.1 |
| J96.0—Acute breathing insufficiency | 24 | 7.3 | 19 | 3.3 | 2.2 |
| J96.9—Unspecified respiratory failure | 16 | 4.8 | 23 | 4.0 | 1.2 |
| **Some conditions originating in the perinatal period (P00-P96)** | **110** | **33.3** | **284** | **49.3** | **0.7** |
| P07.3—Other preterm newborns | 3 | 0.9 | 4 | 0.7 | 1.3 |
| P21.9—Asphyxia at birth unspecified | 7 | 2.1 | 10 | 1.7 | 1.2 |
| P26.9—Unspecified pulmonary hemorrhage originating in the perinatal period | 5 | 1.5 | 9 | 1.6 | 0.9 |
| P28.5—Newborn respiratory failure | 40 | 12.1 | 91 | 15.8 | 0.8 |
| P29.1—Neonatal cardiac dysrhythmia | 4 | 1.2 | 9 | 1.6 | 0.8 |
| P36.9—Unspecified bacterial septicemia of the newborn | 14 | 4.2 | 41 | 7.1 | 0.6 |
| P60—Disseminated intravascular coagulation of the fetus and newborn | 3 | 0.9 | 4 | 0.7 | 1.3 |
| P96.8—Other specified conditions originating in the perinatal period | 6 | 1.8 | 19 | 3.3 | 0.5 |
| **Congenital malformations, chromosomal deformities, and anomalies (Q00-Q99)** | **20** | **6.1** | **43** | **7.5** | **0.8** |
| Q02—Microcephaly | 12 | 3.6 | 1 | 0.2 | 18.0 |
| Q89.7—Multiple congenital malformations not classified elsewhere | 3 | 0.9 | 8 | 1.4 | 0.6 |
| **Abnormal symptoms, signs and findings from clinical and laboratory examinations, unclassified elsewhere (R00-R99)** | **61** | **18.5** | **65** | **11.3** | **1.6** |
| R09.2—Respiratory failure | 8 | 2.4 | 12 | 2.1 | 1.1 |
| R57.0—Cardiogenic shock | 5 | 1.5 | 12 | 2.1 | 0.7 |
| R57.8—Other forms of shock | 9 | 2.7 | 4 | 0.7 | 3.9 |
| R57.9—Unspecified shock | 5 | 1.5 | 7 | 1.2 | 1.3 |
| R68.8—Other specified general symptoms and signs | 15 | 4.5 | 16 | 2.8 | 1.6 |
| R99—Other ill-defined and unspecified causes of mortality | 8 | 2.4 | 3 | 0.5 | 4.8 |
| **Injuries, poisonings, and some other consequences of external causes (S00-T98)** | **10** | **3.0** | **13** | **2.3** | **1.3** |
| T17.9—Foreign body in the respiratory tract unspecified part | 4 | 1.2 | 2 | 0.3 | 4.0 |
| Other groups and types of causes | 18 | 5.5 | 31 | 5.4 | 1.0 |
| **Total** | **330** | **100.0** | **576** | **100.0** | **1.0** |

Source: Center of Data and Knowledge for Health-CIDACS: Linkage of the Live Birth Information System/SINASC. Public Health Events Registry/RESP and Mortality Information System/SIM.

[1]ICD 10 (International Classification of Diseases and Causes of Death (ICD 10th Revision)).

Only causes of death whose absolute frequency were ≥3 are included separately.

PM% calculated in relation to the total of causes of death.

CA, congenital anomalies; CNS, central nervous system; CZS, congenital Zika syndrome; PM, proportional mortality; PMRc, proportional mortality ratio by cause.

**Table 4. Main contributing causes of death (number and proportional mortality/PM%) of 403 children up to 36 months of age, born with CZS, values of these indicators for those born with CA of the CNS non-Zika related and PMRc according to groups and types of causes[1]; Brazil, 2015–2018.**

| Groups and types of causes[1] | CZS | | CA of CNS non-Zika related | | PMRc |
|---|---|---|---|---|---|
| | N | PM(%) | N | PM(%) | |
| **Some infectious and parasitic diseases (A00—B99)** | **21** | **6.0** | **6** | **1.1** | **5.5** |
| A41.9—Unspecified septicemia | 3 | 0.9 | 2 | 0.4 | 2.3 |
| A92.8—Other specified viral fevers transmitted by mosquitoes | 11 | 3.2 | - | - | - |
| **Doenças endócrinas, nutricionais e metabólicas (E00–E90)** | **14** | **4.0** | **4** | **0.7** | **5.7** |
| E43—Severe protein-calorie malnutrition unspecified | 3 | 0.9 | - | - | - |
| E46—Unspecified protein-calorie malnutrition | 6 | 1.7 | 2 | 0.4 | 4.3 |
| **Nervous system diseases (G00-G99)** | **27** | **7.8** | **26** | **4.8** | **1.6** |
| G40.4—Other epilepsies and generalized epileptic syndromes | 3 | 0.9 | - | - | - |
| G40.9—Epilepsy, unspecified | 3 | 0.9 | 2 | 0.4 | 2.3 |
| G80.9—Unspecified cerebral palsy | 3 | 0.9 | 1 | 0.2 | 4.5 |
| G91.9 -Unspecified hydrocephalus | 8 | 2.3 | 11 | 2.0 | 1.2 |
| G93.4—Unspecified encephalopathy | 3 | 0.9 | 2 | 0.4 | 2.3 |
| **Genitourinary system diseases (N00-N99)** | **4** | **1.1** | **7** | **1.3** | **0.8** |
| N17.9—Acute kidney failure unspecified | 3 | 0.9 | - | - | - |
| **Some conditions originating in the perinatal period (P00-P96)** | **77** | **22.1** | **145** | **26.8** | **0.8** |
| P00.2—Fetus and newborn affected by the mother's infectious and parasitic diseases | 10 | 2.9 | 2 | 0.4 | 7.3 |
| P07.0—Very low birth weight newborn | 3 | 0.9 | 6 | 1.1 | 0.8 |
| P07.1—Other low birth weight newborns | 12 | 3.4 | 34 | 6.3 | 0.5 |
| P07.3—Other preterm newborns | 7 | 2.0 | 21 | 3.9 | 0.5 |
| P35.8—Other congenital viral diseases | 4 | 1.1 | - | - | - |
| P36.9—Unspecified bacterial septicemia of the newborn | 3 | 0.9 | 4 | 0.7 | 1.3 |
| P37.1—Congenital toxoplasmosis | 4 | 1.1 | - | - | - |
| P90—Newborn seizures | 3 | 0.9 | 1 | 0.2 | 4.5 |
| **Congenital malformations, chromosomal deformities, and anomalies (Q00-Q99)** | **174** | **50.0** | **305** | **56.4** | **0.9** |
| Q02—Microcephaly | 76 | 21.8 | 3 | 0.6 | 36.3 |
| Q03.1—Atresia of the Clefts of Luschka and the foramen of Magendie | 4 | 1.1 | 4 | 0.7 | 1.6 |
| Q03.9—Unspecified congenital hydrocephalus | 6 | 1.7 | 16 | 3.0 | 0.6 |
| Q04.9—Unspecified congenital malformation of the brain | 3 | 0.9 | 4 | 0.7 | 1.3 |
| Q21.1—Atrial septal defect | 3 | 0.9 | - | - | - |
| Q24.9—Unspecified malformation of the heart | 6 | 1.7 | 8 | 1.5 | 1.1 |
| Q74.3—Multiple congenital arthrogryposis | 3 | 0.9 | - | - | - |
| Q89.7—Multiple congenital malformations, not classified elsewhere | 12 | 3.4 | 11 | 2.0 | 1.7 |
| Q89.9—Unspecified congenital malformations | 11 | 3.2 | 10 | 1.8 | 1.8 |
| Q99.9—Unspecified chromosomal abnormality | 3 | 0.9 | 2 | 0.4 | 2.3 |
| Other groups and types of causes | 31 | 8.9 | 48 | 8.9 | 1.0 |
| **Total** | **348** | **100.0** | **541** | **100.0** | **1.0** |

Source: Center of Data and Knowledge for Health-CIDACS: Linkage of the Live Birth Information System/SINASC. Public Health Events Registry/RESP and Mortality Information System/SIM.

[1]ICD 10 (International Classification of Diseases and Causes of Death (ICD 10th Revision)).

Only causes of death whose absolute frequency were ≥3 are included separately.

PM% calculated in relation to the total of causes of death.

CA, congenital anomalies; CNS, central nervous system; CZS, congenital Zika syndrome; PM, proportional mortality; PMRc, proportional mortality ratio by cause.

perinatal period, namely respiratory system diseases; abnormal symptoms, signs and findings from clinical and laboratory, examinations unclassified elsewhere; and some infectious and parasitic diseases, accounted for the highest number of deaths, with 110, 63, 61, and 41 records, respectively. Newborn respiratory failure (40 deaths; PM = 12.1%), unspecified septicemia (39 deaths; PM = 11.8%), and acute breathing insufficiency (24 deaths; PM = 7.3%) were the most frequently reported immediate causes of death. Microcephaly with PMRc = 18.0, unspecified bronchopneumonia with PMRc = 5.0, other ill-defined and unspecified causes mortality with PMRc = 4.8 contributed to a relative more than 4-fold increase as an immediate cause of death among children with CZS than among children with CNS abnormalities, non-Zika related.

**Contributing causes.** In Table 4, we observe that 348 contributing causes of death were recorded among the children with CZS studied. The ones with the highest number of deaths were the following 174 of the group with congenital malformations, chromosomal deformities, and anomalies; 77 for the group with some conditions originating in the perinatal period; and 27 for the group with nervous system diseases. In absolute numbers, the most frequent recorded contributing causes of death were microcephaly (76; PM = 21.8%), other low birth weight and multiple congenital malformations not classified elsewhere with 12 deaths and PM = 3.4% each. Proportionally, the main contributing causes of deaths among CZS children were microcephaly with PMRc = 36.3, fetus and newborn affected by the mother's infectious and parasitic diseases with PMRc = 7.3, unspecified cerebral palsy and newborn seizures with PMRc = 4.5 each, and unspecified protein-calorie malnutrition with PMRc = 4.3. All these contributing causes represented a more than 4-fold increase in deaths among children with CZS over the comparison group.

Table 5 shows a comparison of the 8 leading underlying causes of death for children with CZS considering these causes anywhere in the sequence of events (rather than underlying only). We observed a considerable increase in the attributable role for unspecified septicemia (5.2-fold), unspecified pneumonia (3.6-fold), and unspecified bacterial septicemia (3.0-fold).

**Table 5. Fold increase in the estimated contribution of leading causes of death recorded for 403 children up to 36 months of age, born with CZS, considering location, anywhere in the causal sequence compared to only using the underlying cause; Brazil, 2015–2018.**

| Leading causes of death/conditions[1] | Only as underlying cause/ condition | | Anywhere in causal sequence | | Fold increase |
|---|---|---|---|---|---|
| | N | % | N | % | |
| Q89.7—Multiple congenital malformations. not classified elsewhere | 18 | 4.5 | 35 | 8.7 | 1.9 |
| A41.9—Unspecified septicemia | 17 | 4.2 | 88 | 21.8 | 5.2 |
| Q04.3—Other deformities due to brain reduction | 12 | 3.0 | 12 | 3.0 | 1.0 |
| P36.9—Unspecified bacterial septicemia of the newborn | 11 | 2.7 | 33 | 8.2 | 3.0 |
| Q89.9—Unspecified congenital malformations | 11 | 2.7 | 28 | 6.9 | 2.5 |
| Q33.6—Lung hypoplasia and dysplasia | 10 | 2.5 | 12 | 3.0 | 1.2 |
| Q00.0—Anencephaly | 10 | 2.5 | 10 | 2.5 | 1.0 |
| Q03.9—Unspecified congenital hydrocephalus | 9 | 2.2 | 18 | 4.5 | 2.0 |
| Q24.9—Unspecified malformation of the heart | 9 | 2.2 | 17 | 4.2 | 1.9 |
| J18.9—Unspecified pneumonia | 8 | 2.0 | 29 | 7.2 | 3.6 |

Conditions that did not constitute a cause of death were not included, even if they were attested as such on the death certificate.

[1]ICD 10 (International Classification of Diseases and Causes of Death (ICD 10th Revision)).

CZS, congenital Zika syndrome.

S1–S4 Tables show the sensitivity analyses using as the comparison group children born before the Zika epidemic in Brazil, with record of CNS anomalies non-Zika-related. The results were similar to the main analyses for most of the causes of death.

## Discussion

In our study, the leading underlying causes of death for the 403 children with CZS were microcephaly, multiple congenital malformations not classified elsewhere, and unspecified septicemia, which together accounted for 31.8% of the total. Among the intermediate causes, the most recorded diseases were microcephaly, unspecified septicemia, and unspecified pneumonia. As immediate causes of death, newborn respiratory failure, unspecified septicemia, and acute respiratory failure were notable as the most recorded. Microcephaly, other low birth weight newborns, and multiple congenital malformations not classified elsewhere were frequently mentioned as contributing causes of death. Children with CZS were more likely to die from most causes mentioned above than children with congenital anomalies of the CNS unrelated to the ZIKV. Although microcephaly is the most frequent diagnosis among reported causes of death for children with CZS, it is not a disease but rather a clinical finding that reflects an underlying CNS pathology, resulting from neurological sequelae that can be multiple and severe [16].

Unspecified septicemia and sepsis are diseases frequently identified in neonates, [17] were the most recorded underlying causes of deaths among the children with CZS up to 36 months of age. Infection and sepsis can be due to several factors that vary by age of the child. In the neonate with CZS, the immature immune system might be further compromised by the frequent occurrence of low and very low birth weight [18,19]. Compromised nutritional status that might have begun before birth can be compounded by neurologic issues such as dysphagia that is common among children with CZS [20–22]. Malnutrition continues to be an issue in a substantial percentage of children with CZS up to 36 months of age, some requiring gastrostomy feeding [20,21] and malnutrition itself can increase vulnerability to infections [19,23]. In the current study, protein-calorie malnutrition was recorded as an underlying and contributing cause of death. Thus, the risk for septicemia can be arise from global issues that are magnified in children with CZS as well as multiple localized sources of infections that are related to the characteristics of this syndrome.

Pneumonia, as well as other diseases of the respiratory system, were recorded very frequently, either as the underlying, intermediate, or immediate cause of death in children with CZS analyzed here. These can be initiated in several ways. For example, the ability of the children with CZS to feed may be impaired by oral motor dysfunction, dysphagia, regurgitation, as well as oral morphological abnormalities [20]. Frequent choking can lead to bronchoaspiration of food and saliva and resultant pneumonia [24,25]. In addition seizures, common in CZS [26], can result in bronchoaspiration of gastric contents and lead to pneumonia or, ultimately, respiratory failure. Pneumonitis due to food or vomiting and other and unspecified seizures were recorded as intermediate causes of death among children with CZS. Other factors that increase the risk of early death in CZS include phrenic nerve palsy with diaphragmatic paralysis that leads to respiratory failure in early infancy [27,28].

Neurogenic bladder, also seen in children with CZS [29], can lead to relapsing urinary infections, and in the absence of adequate clinical management, such infections can be a focus of sepsis and cause the death. Thus, it is not surprising that sepsis, respiratory failure, and newborn respiratory failure are among the leading immediate causes of death in children with CZS.

The motor sequelae of CZS have been linked to spastic or mixed cerebral palsy [30]. Cerebral palsy, low cognitive development, and visual impairment and hearing loss can negatively

impact the child's quality of life, physical well-being, and autonomy for the most basic activities, which further increases dependence on a caregiver [31]. To our knowledge, most children with CZS are born into impoverished families [32] and access to medical support and rehabilitation services might be limited. Without adequate management of the numerous health problems associated with CZS early mortality can result.

Our understanding of the impacts of CZS on the developing fetus is still evolving. These study findings on the circumstances involved in the deaths of these children not only reiterate known aspects of this syndrome as, for example, their greater risk of premature death, but also to which fatal diseases or disease processes they are most vulnerable.

This is the first study to analyze the sequence of morbid events reported in the DC as the underlying, intermediate, immediate, and contributing causes involved in determining deaths among children with CZS up to the age of 36 months. We also compared our findings to a group of children with congenital CNS abnormalities that were non-Zika related. Although not identifying the presence of a specific life-threatening morbidity pattern for children with CZS, this comparison showed that these children were vulnerable to a wide range of causes of death than those in the comparison group. In addition, we performed a sensitivity analysis using a group of children with CNS abnormalities not related to Zika from years before the Zika epidemic. Although the results were similar to the main analyses for most of the causes of death, the size of this group from these previous years were much smaller than during the Zika epidemic years. Another strength of this study was that it captured all the cases of CZS registered in Brazil from 2015 to 2018.

There are, however, limitations. First, the present study was based on registry data, and further clinical information was not available. Second, underreporting in the RESP is possible, mainly among those fetuses prenatally exposed to ZIKV but without detectable malformations immediately after birth. Therefore, misclassification is possible. To overcome this limitation, we performed a sensitivity analysis using a comparison group of children before the Zika epidemic. Third, because we used the final classification provided by the Brazilian Ministry of Health, some degree of misclassification of CZS might have happened. Fourth, the linkage process could have introduced misclassification due to a linkage error when children with CZS were not linked with death records, and therefore, this could also contribute to the underreporting of deaths. There were also laboratory limitations for case confirmation and the use of various confirmation criteria, especially at the beginning of the epidemic of this syndrome, in addition to errors in completing the causes of death on the DC. Regarding this latter limitation, it became evident that physicians should have been alerted about the relevance of correctly filling out this part of the DC, as well as the need for the Epidemiological Surveillance team to adopt measures to adjust the programming of the automatic selector to current recommendations for selecting the underlying cause of death of children with CZS [15], considering that it was possibly programmed to select microcephaly as the cause of death. We highlight the inconsistent registering of causes of death into the various categories. For example, congenital anomalies were listed as underlying, intermediate, immediate, and contributing causes of death and, in particular, the assignment of microcephaly as a cause of death. Another issue is misattribution of certain birth defects as Zika related. Although included among Zika-related birth defects early in the epidemic, subsequent data collection has shown that anencephaly and other non-communicable diseases are not related to congenital Zika infection [33].

Despite these limitations, our study highlighted valuable information on deaths among CZS children by age of 36 months and the importance of preventing ZIKV infection during pregnancy. Although CZS was declared a public health emergency, it was a rare event considering the total number of births in Brazil, and therefore, the information analyzed here was made possible through the linking of the national administrative bases. The presented results

might help develop evidence-based health protocols and generate awareness about the conditions associated with high mortality to guide health care providers responsible for the care of children affected by congenital ZIKV infection.

## Supporting information

**S1 RECORD checklist. The RECORD statement—Checklist of items.**
(DOCX)

**S1 Fig. International model for the medical certificate of cause of death.**
(TIF)

**S2 Fig. Flowchart of the procedure performed by Ministry of Health for the inclusion of causes of death in the Brazilian Mortality Information System (SIM) and the procedure used to select causes of death of live born infants with CZS, up to 36 months of age.** Brazil, 2015 to 2018.
(TIF)

**S3 Fig. Flowchart of the selection process of the deaths of children up to 36 months of age born with CSZ and those born with congenital anomalies of the CNS non-Zika–related for the study of causes of death in children with CZS.** Brazil, 2015 to 2018.
(TIF)

**S1 Table. Main underlying causes of death (number and proportional mortality/PM%) of 403 children up to 36 months of age, born with CZS 2015–2018, values of these indicators for those born with CA of the CNS non-Zika related, 2012–2013 and proportional mortality ratio between causes (PMRc) according to groups and types of causes in Brazil.**
(DOCX)

**S2 Table. Main intermediate causes of death (number and proportional mortality/PM%) of 403 children up to 36 months of age, born with CZS 2015–2018, values of these indicators for those born with CA of the CNS non-Zika related, 2012–2013 and proportional mortality ratio between causes (PMRc) according to groups and types of causes in Brazil.**
(DOCX)

**S3 Table. Main immediate causes of death (number and proportional mortality/PM%) of 403 children up to 36 months of age, born with CZS 2015–2018, values of these indicators for those born with CA of the CNS non-Zika related, 2012–2013 and proportional mortality ratio between causes (PMRc) according to groups and types of causes in Brazil.**
(DOCX)

**S4 Table. Main contributing causes of death (number and proportional mortality/PM%) of 403 children up to 36 months of age, born with CZS 2015–2018, values of these indicators for those born with CA of the CNS non-Zika related, 2012–2013 and proportional mortality ratio between causes (PMRc) according to groups and types of causes in Brazil.**
(DOCX)

## Acknowledgments

We thank the data production team CIDACS/FIOCRUZ collaborators for their work linking these data and the Information Technology team for enabling access to data. We also thank the technical team of the Municipal Health Department, Department of Health Information, Salvador, Bahia, Brazil, especially João Batista Vieira Figueredo, Joselito Ramos de Andrade,

and Igor Bittencourt Santos for the clarifications on the selection process of the causes of death that constituted the causal sequence of death of the children included in our study.

## Author Contributions

**Conceptualization:** Maria da Conceição N. Costa, Maria Gloria Teixeira, Enny S. Paixão.

**Formal analysis:** Luciana Lobato Cardim, Eliene dos Santos de Jesus.

**Funding acquisition:** Mauricio L. Barreto, Laura C. Rodrigues, Liam Smeeth, Enny S. Paixão.

**Methodology:** Maria da Conceição N. Costa.

**Supervision:** Maria da Conceição N. Costa, Cynthia A. Moore, Enny S. Paixão.

**Validation:** Enny S. Paixão.

**Writing – original draft:** Maria da Conceição N. Costa, Luciana Lobato Cardim, Rita Carvalho-Sauer, Enny S. Paixão.

**Writing – review & editing:** Maria da Conceição N. Costa, Luciana Lobato Cardim, Cynthia A. Moore, Eliene dos Santos de Jesus, Mauricio L. Barreto, Laura C. Rodrigues, Liam Smeeth, Lavínia Schuler-Faccini, Elizabeth B. Brickley, Wanderson K. Oliveira, Eduardo Hage Carmo, Julia Moreira Pescarini, Roberto F. S. Andrade, Moreno M. S. Rodrigues, Rafael V. Veiga, Larissa C. Costa, Giovanny V. A. França, Maria Gloria Teixeira, Enny S. Paixão.

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
