## [Editor Report · Decision Letter 0]

5 Aug 2022

Dear Dr Costa, 

Thank you for submitting your manuscript entitled "Causes of Death in Children with Congenital Zika Syndrome: A nationwide cross-sectional study. Brazil, 2015-2018." for consideration by PLOS Medicine.

Your manuscript has now been evaluated by the PLOS Medicine editorial staff and I am writing to let you know that we would like to send your submission out for external peer review.

Please re-submit your manuscript within two working days, i.e. by Aug 09 2022 11:59PM.

Kind regards,

Caitlin Moyer, Ph.D.

Associate Editor

PLOS Medicine

---

## [Decision Letter · Decision Letter 1]

31 Oct 2022

Dear Dr. Paixao,

Thank you very much for submitting your manuscript "Causes of Death in Children with Congenital Zika Syndrome: A nationwide cross-sectional study. Brazil, 2015-2018." (PMEDICINE-D-22-02611R1) for consideration at PLOS Medicine. 

Your paper was evaluated by an associate editor and discussed among all the editors here. It was also discussed with an academic editor with relevant expertise, and sent to independent reviewers, including a statistical reviewer. The reviews are appended at the bottom of this email and any accompanying reviewer attachments can be seen via the link below:

[LINK]

In light of these reviews, I am afraid that we will not be able to accept the manuscript for publication in the journal in its current form, but we would like to consider a revised version that addresses the reviewers' and editors' comments. Obviously we cannot make any decision about publication until we have seen the revised manuscript and your response, and we plan to seek re-review by one or more of the reviewers. 

We hope to receive your revised manuscript by Nov 21 2022 11:59PM. Please email us (plosmedicine@plos.org) if you have any questions or concerns.

We look forward to receiving your revised manuscript. 

Sincerely,

Callam Davidson

Associate Editor

PLOS Medicine

plosmedicine.org

Please rearrange your title such that it reads: ‘Causes of Death in Children with Congenital Zika Syndrome in Brazil, 2015-2018.: A nationwide cross-sectional study.’

Please structure your abstract using the PLOS Medicine headings (Background, Methods and Findings, Conclusions).

Abstract Methods and Findings:

* Please ensure that all numbers presented in the abstract are present and identical to numbers presented in the main manuscript text.

Please ensure that the study is reported according to the RECORD guideline, and include the completed RECORD checklist as Supporting Information. Please add the following statement, or similar, to the Methods: "This study is reported as per the REporting of studies Conducted using Observational Routinely-collected health Data (RECORD) Statement (S1 Checklist)."

The RECORD guideline can be found here: https://www.equator-network.org/reporting-guidelines/record/

Did your study have a prospective protocol or analysis plan? Please state this (either way) early in the Methods section.

Supporting Information should be uploaded separately rather than embedded in the main text (e.g., Figure S1-S3). In the event of publication, the citation will be hyperlinked. 

The ‘Contributors’, ‘Declarations’ and ‘Funding’ sections can all be removed from the end of the main text – this information is captured via the Submission Form.

References: Journal name abbreviations should be those found in the National Center for Biotechnology Information (NCBI) databases.

Comments from the reviewers:

Reviewer #1: Thanks for the opportunity to review your manuscript. My role is as a statistical reviewer, so my review concentrates on the study design, data, and analysis that are presented. I have put general questions first, followed by queries relevant to a specific section of the manuscript (with a page/line reference).

The study used linked population health data (births, deaths, notifications of potential abnormalities from ZIKV). The linkage was accomplished with a probabilistic linkage tool, with information available about the tool and the performance of the tool is publically available. The underlying datasets for the study are well-explained, thank you. The key analysis is in a decedent cohort of composed of children who died of a) Zika congenital anomalies and b) children who died of non-Zika CNS congenital abnormalities. The relative rate (ratio) of proportionate mortality (% of those who died of particular condition within exposure group) between a ) and b) is used to identify causes of death seen more (or less) frequently in children who died of Zika compared to children who died to other CNS congenital abnormalities. This effect measure is typically used where underlying population for each exposure is not or cannot be measured. 

Was any information about the children (particularly age distribution) in Zika vs. non-Zika available? This would be helpful to include in the manuscript.

I would add confidence intervals for PMRC (proportional mortality ratio by cause) where a cause of death category for both CZS and the non-Zika has at least 5 deaths in each. Similarly for the calculation of rate of cause of death as underlying cause vs. anywhere in causal sequence. I would also consider reformulating the data used to calculate mortality odds ratio instead of PMRC (Am J Epidemiol 1981; 114: 144-148.), it can avoid bias when there are influences on the underlying denominators. 

P5, L135. Are these type of congenital abnormalities (or congenital abnormalities) a notifiable disease in Brazil (i.e. legally required to report any incidence)? I had a look at the sites in the references but google translate did not seem to do a good job translating Portuguese to English. 

P12, L269. I would consider dropping the 'chapter' column here 

Reviewer #2: The manuscript by Costa and colleagues analyzes the causes of death as well as the probably sequence of morbid events leading to death in children in the first 36 months of life in children born with CZS in Brazil from 2105 to 2018. They compare the causes of death in this group to causes of death in a group of children born during the same time period who had congenital neurologic abnormalities which were not attributable to maternal Zika virus infection. The source of data included Brazilian public health databases including SINASC which reports liver births, RESP which reports public health event records and maintains a database of congenital infections and SIM, which is a mortality information database. The authors collected data on all children who had CZS and who died by 36 months of age and compared the causes of death and the sequence of events leading to death as abstracted from death certificates with causes of death reported for children who were born with neurologic abnormalities during the same period but per classification in the health information system did not have congenital Zika syndrome. The authors conclude that the causes of death were slightly different in both groups, and that children with CZS more often died of severe protein-caloric malnutrition, unspecified pneumonia and asphyxia at birth than those in the comparison group. 

One of the concerns with this approach to the analysis is that there is a potential problem with the comparator group in that these children were also born during the time of the Zika virus epidemic in Brazil. Congenital Zika virus has unique features, however the laboratory diagnosis of congenital infection is not straightforward. The virus may not be present in all infants at birth, while many infants might not have been tested. Maternal infection may not have a laboratory diagnosis, the window period for viral detection is short, other arboviral infections may cause a rash. Chikungunya for example circulated in some parts of Brazil concurrently with ZIKV in some states. It is nearly impossible to ascertain that the children selected with the Zika unrelated neurologic conditions may not have had CZS as well. How can one be sure that the CNS abnormalities of the control group were definitely not due to Zika if these children born during the Zika epidemic period in Brazil? CZS causes a myriad of CNS findings and without knowing the specific details, as for example, was there a confirmed genetic diagnosis, it is practically impossible to ascertain whether the CNS abnormalities in the non-Zika group were truly independent of maternal ZIKV infection. It does not appear that they were recruited prior to the pandemic, they seem to be a simultaneous cohort. Several cases might have been misclassified which the authors acknowledge in the discussion. Code Q00 corresponds to neural tube defects, Q01 to encephalocele, Q05 spina bifida, Q06 other congenital malformations of the spinal cord, Q07 other congenital malformations of the CNS. Any of these could occur in children with CZS; lack of linkage to a Zika etiology in a database does not necessarily mean that Zika was not behind it. Zika causes a spectrum of neurologic manifestations and all of the findings described in the non-Zika cohort can be found in children with Zika. A comparator group of children born during the pandemic should not be used given the difficulties in establishing a definitive diagnosis. A pre Zika epidemic comparator group would be more suitable.

However, what the comparator group adds to the analysis is not clear cut, In addition to being nearly impossible to ascertain that the non-Zika group did not have ZIKV infection, if children have very similar deformities, the repercussions of these deformities are likely the same. If one compares more serious incapacitating CNS lesions with others that are less incapacitating, certainly the proportional mortality will be higher in those with more serious manifestations of disease. So this is somewhat of comparison between apples and oranges which is not very helpful. It would be more interesting for the authors to focus on children in the Zika cohort only and show what factors distinguished children with CZS who died by 36 months of age with children with CZS who did not die by this age group. This would be more helpful clinically because it would highlight areas that medical providers need to focus on. They could continue to describe the cascade of diagnoses that led to death but without trying to compare it to the other group, but instead, showing what features were risk factors for death by 36 months in comparison to children with CZS who did not die. The data is available in their database. The other option is to use a comparator group of the same age but recruited before the Zika epidemic, however an analysis of risk factors leading to death in CZS children only and then the inclusion of the sequence of events that led to death in CZS children would be more informative to readers.

Another important consideration is that there are some children with CZS who appear to have other diagnoses which could lead to the CNS findings reported. For example there are 5 children with congenital toxoplasmosis reported in Table 1 in the CZS group. If children have a diagnosis of congenital toxoplasmosis, another congenital infection, they should be excluded from the CZS group because their neurologic manifestations can be attributed to another cause. In the same way, any child with a diagnosed genetic defect or chromosomal abnormality should not be part of the CZS group, these should be excluded. 

Specific comments: Should proofread the manuscript for English corrections.

Abstract. 

Line 46: damage to the…

Line 47, comma should be inserted after CNS

The abstract should mention causes of death were abstracted from death certificates using the WHO death certificate template.

Abstract should make it clear that the inclusion criteria for both CZS and non-CZS children was death by 36 months of age, because it is not clear. 

Line 62: Fetuses 

Line 65: Delete those

Line 66: to the group born with, Also, delete the duplicate with

Conclusion of the abstract: not clearly supported by the results provided since data for the comparator group is not provided in the abstract and there are no statistical analyses included to demonstrate that they were significantly different. 

Introduction.

Line 98: Would not say all but most. There are problems in congenital Zika syndrome not necessarily related to central or peripheral nervous system damage, such as infants being born small for gestational age (SGA), or a higher prevalence of congenital heart defects both are described in children exposed to maternal ZIKV Infection in utero.

Line 111: up to 36 months of age instead of by

Line 111: delete occurred

Line 130-131: who assisted in the delivery

Line 131: As early as

Line 171: are instead of is

Line 172: are coded with the appropriate ICD-10

Line 173: inputted instead of inserted

Figure S2: Typos include: 

*5- line d if DE were filled instead of well. Also where only one line was filled instead of were.

*6- after automated SIM were considered other types of causes? The sentence does not make sense as written. 

Line 183: Flowchart of the procedures performed instead of indicated 

Results:

Line 248: Would state that 36.6% of the children were dead by 3 years of age, 403 of 1110. 

Line 252: 2434 of how many? Per Figure 3 the denominator would be 19197, or 12.7%.

Discussion:

Line 398-399: Children with CZS may be born small for gestational age, which contributes to the malnutrition. They may have seizures which can also lead to aspiration pneumonia, in addition to dysphagia. 

Reviewer #3: This is a descriptive study of the causes of death in children with Congenital Zika Syndrome. It is an unambitious study and based on secondary data from health information systems, therefore, subject to problems related to the criteria for defining cause of death and quality of filling out death certificates. Even so, I think it is important to be published as a way of documenting the findings related to the country most affected by this disease. Unless I am mistaken, this is the largest number of deaths associated with the Zika Virus ever published and therefore a document of historical value and will allow monitoring of CZS cases and their complications.

[LINK]

---

## [Decision Letter · Decision Letter 2]

12 Jan 2023

Dear Dr. Paixao,

Thank you very much for re-submitting your manuscript "Causes of Death in Children with Congenital Zika Syndrome in Brazil, 2015-2018.: A nationwide Record Linkage Study." (PMEDICINE-D-22-02611R2) for review by PLOS Medicine.

I have discussed the paper with my colleagues and the academic editor and it was also seen again by two reviewers. I am pleased to say that provided the remaining editorial and production issues are dealt with we are planning to accept the paper for publication in the journal.

[LINK]

We look forward to receiving the revised manuscript by Jan 19 2023 11:59PM.   

Sincerely,

Callam Davidson, 

Senior Editor 

PLOS Medicine

plosmedicine.org

Requests from Editors:

Abstract Conclusions: Please avoid vague statements such as "reflects important aspects of CZS”. Mention only specific implications substantiated by the results.

Please trim the 7th bullet point in your Author Summary – ideally, bullet points should only run to 1-2 sentences.

Line 153: Please combine these citations (i.e., [2, 3]).

Line 179: I could not locate the RECORD checklist in the Supporting Information, please include the completed checklist (when completing the checklist, please use section headings and paragraph numbers, rather than line/page numbers).

Please ensure you have defined all abbreviations in Table and Figure legends (including in the Supporting Information).

Lines 266-271: This sentence is both long and quite confusing – please check for clarity.

Line 284: Sensitivity analysis.

Please confirm that no informed consent was required for this study. 

Please ensure that any changes in the analysis-- including those made in response to peer review comments-- are identified as such in the Methods section of the paper, with rationale.

The legends of Tables 1-4 and S1-S4 contain a flag (1) but I could not locate the corresponding flag in the tables.

Lines 452-454: This sentence does not scan properly, please confirm there are no omissions. 

Line 496: Please temper claims to primacy by including ‘to our knowledge’, or similar. 

Reference 4 contains a flag (†), please delete. 

Line 420: Please report Results in the past tense (‘we observed’).

Comments from Reviewers:

Reviewer #1: Thanks for the revised manuscript and responses to my review. I agree that CIs are not necessary given it's a study of an entire population. I follow the analysis presented in Table 5 better now, this makes sense to me in this revision of the manuscript. The updates to the manuscript resolve the rest of my original queries. 

Reviewer #2: The authors have successfully addressed the questions raised in the prior review.

[LINK]

---

## [Editor Report · Decision Letter 3]

23 Jan 2023

Dear Dr Paixao, 

On behalf of my colleagues and the Academic Editor, Dr Rebecca Grais , I am pleased to inform you that we have agreed to publish your manuscript "Causes of Death in Children with Congenital Zika Syndrome in Brazil, 2015-2018.: A nationwide Record Linkage Study." (PMEDICINE-D-22-02611R3) in PLOS Medicine.

PRESS

Sincerely, 

Callam Davidson 

Associate Editor 

PLOS Medicine